# Exploring the Association of Burning Mouth Syndrome with Depressive and Anxiety Disorders in Middle-Aged and Older Adults: A Systematic Review

**DOI:** 10.3390/jpm13061014

**Published:** 2023-06-19

**Authors:** Vittorio Dibello, Andrea Ballini, Madia Lozupone, Carlo Custodero, Stefania Cantore, Rodolfo Sardone, Antonio Dibello, Filippo Santarcangelo, Bianca Barulli Kofler, Massimo Petruzzi, Antonio Daniele, Vincenzo Solfrizzi, Francesco Panza

**Affiliations:** 1Department of Orofacial Pain and Dysfunction, Academic Centre for Dentistry Amsterdam (ACTA), University of Amsterdam and Vrije Universiteit Amsterdam, 1081 HV Amsterdam, The Netherlands; 2“Cesare Frugoni” Internal and Geriatric Medicine and Memory Unit, University of Bari Aldo Moro, 70124 Bari, Italy; carlo.custodero@uniba.it (C.C.); bianca.barulli@gmail.com (B.B.K.); vincenzo.solfrizzi@uniba.it (V.S.); 3Department of Clinical and Experimental Medicine, University of Foggia, 71122 Foggia, Italy; andrea.ballini@me.com; 4Department of Precision Medicine, University of Campania “Luigi Vanvitelli”, 80138 Naples, Italy; stefaniacantore@pec.omceo.bari.it; 5Neurodegenerative Disease Unit, Department of D-BRAIN, University of Bari Aldo Moro, 70124 Bari, Italy; madia.lozupone@gmail.com; 6Independent Researcher, 70124 Bari, Italy; filipposantarcangelo@gmail.com; 7Unit of Research Methodology and Data Sciences for Population Health, National Institute of Gastroenterology “Saverio de Bellis”, Research Hospital, Castellana Grotte, 70013 Bari, Italy; rodolfo.sardone@uniba.it; 8Stella Maris Nursing Home and Day Care Center, Monopoli, 70043 Bari, Italy; antonio.dibello@asl.bari.it; 9Interdisciplinary Department of Medicine, Section of Dentistry, University of Bari Aldo Moro, 70124 Bari, Italy; massimo.petruzzi@uniba.it; 10Institute of Neurology, Catholic University of Sacred Heart, 00168 Rome, Italy; antonio.daniele@unicatt.it; 11Institute of Neurology, Fondazione Policlinico Universitario A. Gemelli IRCCS, 00168 Rome, Italy

**Keywords:** depression, anxiety, burning mouth syndrome, older people, middle age

## Abstract

Background: Burning Mouth Syndrome (BMS) is an idiopathic condition mainly affecting middle-aged and older individuals with hormonal disturbances or psychiatric disorders and is characterized by chronic pain. The etiopathogenesis of this multifactorial syndrome is largely unknown. The objective of the present systematic review was therefore to evaluate the relationship of BMS with depressive and anxiety disorders in middle-aged and older individuals. Methods: We selected studies evaluating BMS and depressive and anxiety disorders assessed with validated tools, published from their inception up to April 2023, using PubMed, MEDLINE, EMBASE, Scopus, Ovid, and Google Scholar databases and adhering to the PRISMA 2020 guidelines/PRISMA 2020 27-item checklist. This study is registered on PROSPERO (CRD42023409595). The National Institutes of Health Quality Assessment Toolkits for Observational Cohort and Cross-Sectional Studies were used to examine the risk of bias. Results: Two independent investigators rated 4322 records against the primary endpoint and found 7 records meeting the eligibility requirements. Anxiety disorders were found to be the most common psychiatric disorders related to BMS (63.7%), followed by depressive disorders (36.3%). We found a moderate association of BMS with anxiety disorders, with multiple studies included (*n* = 7). Moreover, we found a low association of BMS with depressive disorders (included studies, *n* = 4). The role of pain appeared to be controversial in explaining these associations. Conclusions: In middle-aged and older subjects, anxiety and depressive disorders may be potentially related to the development of BMS. Furthermore, also in these age groups, females showed higher risk of developing BMS than males, even when taking into account multimorbidity such as sleep disorders, personality traits, and biopsychosocial changes as suggested by study-specific findings.

## 1. Introduction

Burning mouth syndrome (BMS) is a multifactorial and idiopathic syndrome characterized by a persevering burning sensation and chronic pain in clinically normal oral mucosa without blood test abnormalities [1,2]. In clinical practice, stomatodynia (mouth pain), stomatopyrosis (mouth burning), glossodynia (tongue pain), glossopyrosis (tongue burning), and oral dysesthesia are some terms used for identifying BMS [2]. The intensity of pain symptoms determines the classification of BMS as mild, moderate, and severe, although the majority of patients with this syndrome may experience moderate to severe burning sensations [mean severity: about 5–8 cm on a 0–10 cm visual analog scale (VAS)] [3]. Scala and colleagues, in 2003, suggested distinguishing this syndrome from “primary BMS,” which is essentially an idiopathic form of this condition, and “secondary BMS,” which results from a systemic or local pathological condition [2]. In 2020, the International Headache Society categorizes BMS as neuropathic and facial pains in the first edition of the International Classification of Orofacial Pain, describing this condition as “an intraoral burning or dysaesthetic sensation, recurring daily for more than 2 h per day for more than 3 months, without evident causative lesions on clinical examination and investigation” [4].

Due to the non-specific nature of the ailments of BMS, there is little information on the clinical picture of this syndrome. The oral-pain pattern of BMS and its relationship with local, systemic, and/or psychogenic disorders has suggested a further distinction in three subtypes [5], although without a confirmed validity. In particular, Types 1 and 3 BMS, clearly linked to local or systemic factors, do not provoke sleep disorders [2]. On the contrary, individuals with Type 2 BMS, with persistent daily symptoms, depressive and anxiety disorders, and a reduced desire for socialization, reported habitually also an altered sleep pattern [2]. According to an epidemiologic study, the prevalence of BMS is estimated to be approximately 5% in the general population, and it is mainly observed in middle-aged or older adult post-menopausal women [6]. Psychobiological risk factors for the development of BMS may include low education levels; cerebrovascular disorders and stroke; depressive, anxiety, and personality disorders; vitamin deficiencies; stressful life events; excessive use of hexetidine mouthwashes; and drugs, i.e., angiotensin-converting enzyme inhibitors and anticoagulants [6]. Moreover, in Parkinson’s disease, BMS is also frequent, characterized by dopamine dysregulation, confirmed by positron emission tomography (PET) in the nigrostriatal dopaminergic pathway [7].

In accordance with the biopsychosocial model of chronic pain as a multidimensional phenomenon, in which psychological, cognitive, and emotional factors, as well as multimorbidity, may play a role in pain [8], depressive and anxiety disorders are the most common and the most frequently investigated psychiatric conditions in BMS patients [9,10]. While there are some systematic reviews and meta-analyses investigating the relationship of depressive and anxiety disorders in BMS patients without selection for age groups [11,12], a subgroup analysis by age in a large and recent meta-analysis showed the prevalence was higher for individuals over 50 years (3.31%) than under 50 years (1.92%) [13]. Therefore, the objective of the present systematic review was to investigate the relationship of BMS with depressive and anxiety disorders in middle-aged and older adults.

## 2. Materials and Methods

### 2.1. Search Strategy and Data Extraction

The Preferred Reporting Items for Systematic Reviews and Meta-Analyses (PRISMA) 2020 guidelines, adhering to the PRISMA 2020 27-item checklist [14], were followed to perform the present systematic review. Separate searches were performed in the EMBASE, Scopus, Ovid, Google Scholar, US National Library of Medicine (PubMed), and Medical Literature Analysis and Retrieval System Online (MEDLINE) databases to retrieve original articles exploring associations between BMS (exposure) and depressive and anxiety disorders (outcome). Inclusion criteria required that the exposure factors were selected to include any study assessing BMS, both clinically and through instrumental and laboratory investigations, while for the outcome(s), which referred to depressive and anxiety disorders, we selected only studies using widely accepted and validated clinical criteria. The age of 40 years or older was also an inclusion criterion applied when skimming for original studies correlating BMS diagnoses and depressive and anxiety disorders. Among exclusion criteria, we did not include technical reports, letters to the editor, and systematic and narrative review articles. No skimming was applied to the recruitment settings (home care, hospital, community) or general health status of subjects. In Appendix A, we show the search strategy used in PubMed and adapted for the other four electronic databases. The search strategy covers the timeframe from the database creation to 2 April 2023 without language limitations. Two different investigators (VD, AB) searched for original articles, screening abstracts and titles of the retrieved papers separately and in duplicate, checking the complete texts, and selecting records.

### 2.2. Protocol and Registration

For the present systematic review, we established and registered a priori protocol on PROSPERO, a prospective international register of systematic reviews (CRD42023409595). The two investigators (VD, AB) separately and in duplicate extracted the following information in a piloted form: (1) some general information (author, year of publication, design, settings, country, sample size, and age), (2) two different subtypes of psychological disorders (namely depression and anxiety), and (3) outcome assessment tools (different questions from validated questionnaires). We managed all selected original articles with the MS Excel software (Microsoft Office Ver n. 365) platform for data collection, excluding all duplicated records. All data were cross-checked, discrepancies were discussed, and disagreements were resolved by a third investigator (FP).

### 2.3. Quality Assessment within and across Studies and Overall Quality Assessment

Paired investigators (VD and AB or ML) independently appraised the methodological quality of selected studies using the National Institutes of Health Quality Assessment Toolkits for Observational Cohort and Cross-Sectional Studies [15]. According to the criteria included in the toolkit, we assigned the ratings high (good), moderate (fair), or low (poor) to included studies. The toolkit included 14 items as evaluating factors associated with type I and type II errors, the risk of bias, transparency, and confounding factors (population, study question, participation rate, sample size justification, inclusion criteria, time frame, time of measurement of exposure/outcomes, levels of the exposure, defined exposure, blinded assessors, repeated exposure, defined outcomes, loss to follow-up, and confounding factors). The maximum possible scores for cross-sectional and prospective studies were 8 and 14, respectively, given that items 6, 7, and 13 do not refer to cross-sectional studies. A fourth investigator (FP) resolved possible disagreements regarding the methodological quality of the included studies. For assessing the overall quality of evidence in the included studies, we used a modified version of the Grading of Recommendations Assessment, Development and Evaluation (GRADE) rating system, considering the following factors: the strength of association for BMS and depressive and anxiety disorders, methodological quality/design of the included studies, consistency, directedness, precision, size, and (where possible) dose-response gradient of the estimates of effects across the evidence base. As with a GRADE rating system, we graded evidence as very low, low, moderate, and high.

## 3. Results

After a preliminary systematic search of the literature, we yielded 4322 records, and, excluding duplicates, we considered 228 records potentially relevant after the title and abstract analysis. At this stage, we excluded 51 records for not meeting the requirements of the present review target. After examining the full text of the remaining 177 records, we included only 7 articles meeting the inclusion criteria in the final qualitative analysis. In Figure 1, we show the Preferred Reporting Items for Systematic Reviews and Meta-analyses (PRISMA) flow chart indicating the number of studies at each stage of the review.

Our literary skimming process retrieved seven eligible articles [1,16,17,18,19,20,21]. Table 1 shows the study design, sample size (N) and gender ratio (%), minimum age and mean (standard deviation, SD), setting, and country of included studies. For all selected studies, a hospital outpatient setting was found (100%, *n* = 7). The European continent led the geographical distribution of included studies (42.8%, *n* = 3, of which 2 from Italy and one from Sweden), followed by Asia (28.6%, *n* = 2), in addition to South America (28.6%, *n* = 2). This last finding suggested heterogeneity in geographic distribution and inadequate cross-country representativeness. Among 1615 subjects, the majority were females (85% versus 15%). All selected studies were case control, among which, however, cross-sectional design (57.1%, *n* = 4) was more common than prospective cohort (28.6%, *n* = 2) or retrospective cohort (14.3%, *n* = 1).

### 3.1. Depressive and Anxiety Disorders and Their Distribution across Different Studies

In Figure 2, we show the percentage distribution of the investigated psychiatric disorders.

Given the multiple outcomes found in 4 of the 7 selected studies, a total of 11 outcomes were recorded as denominators when calculating the representativeness of each psychiatric outcome. More specifically, four studies were found to evaluate two different outcomes each [1,16,19,20]. Overall, anxiety disorders were found to be the most common (63.7%, *n* = 7 out of 11), followed by depressive disorders (36.3%, *n* = 4 out of 11).

### 3.2. Assessment Tools of Depressive Disorders and Their Distribution across Different Studies

Among the assessment tools investigating depressive disorders, the Hamilton Rating Scale for Depression (HAM-D) (28.5%, *n* = 2) was the most frequently used tool, followed by the Hospital Anxiety and Depression (HAD) Scale (14.3%, *n* = 1), the Mini-International Neuropsychiatric Interview-Plus (MINI-Plus) (14.3% *n* = 1), the Beck Depression inventory (BDI) (14.3%, *n* = 1), the Symptom Checklist-90-Revised (SCL-90-R) (14.3%, *n* = 1), and the Korean Standard Classification of Diseases, Sixth Revision (KCD-6) (14.3%, *n* = 1).

### 3.3. Assessment Tools of Anxiety Disorders and Their Distribution across Different Studies

Among the assessment tools investigating anxiety disorders, the State-Trait Anxiety Inventory (STAI) (22.3%, *n* = 2) was the most frequently used diagnostic instrument, followed by the Hospital Anxiety and Depression (HAD) Scale (11.1%, *n* = 1), the Beck Anxiety Inventory (BAI) (11.1%, *n* = 1), the Cattell Anxiety Scale (11.1%, *n* = 1), the Mini-International Neuropsychiatric Interview-Plus (MINI-Plus) (11.1% *n* = 1), the Symptom Checklist-90-Revised (SCL-90-R) (11.1%, *n* = 1), the Korean Standard Classification of Diseases, Sixth Revision (KCD-6) (11.1%, *n* = 1), and the Swedish universities Scales of Personality (SSP) (11.1%, *n* = 1).

### 3.4. Summary of Findings on the Association of Burning Mouth Syndrome with Depressive and Anxiety Disorders in Middle-Aged and Older Adults

Examining the case-control studies in middle-aged and older adults selected for the present systematic review, we found a moderate association of BMS with anxiety disorders, with multiple studies included (*n* = 7), with a large sample size (*n* = 1.615), and with partly provided estimates [hazard ratio (HR) from 1.72 to 2.13, 95% confidence intervals (CI) from 1.30 to 1.57 (lower) and from 2.29 to 2.91 (higher) and an odds ratio (OR) of 4.26, 95% CI = 1.78–10.15]. Moreover, we found a low association of BMS with depressive disorders, with a few studies included (*n* = 4), with a large sample size (*n* = 1.343), and with partly provided estimates [HR from 1.45 to 1.68, 95% CI from 1.03 to 1.15 (lower) and from 2.04 to 2.43 (higher) and an OR of 3.83, 95% CI = 1.53–9.57]. In particular, study-specific findings showed that BMS was more present in subjects with anxiety disorders (63.7%) than in those with depressive disorders (36.3%). Furthermore, females showed higher risk of developing BMS than males also of middle and older age (seven of seven studies). Anxiety and depressive disorders in subjects with BMS did not appear to be correlated and aggravated by the intensity of pain (two of seven studies), but depressive symptoms could contribute to pain (one of seven studies). Finally, there was a worsening of symptoms such as sleep disorders, personality traits, and biopsychosocial changes (three of seven studies), in addition to anxiety and depression in subjects with BMS (Table 1 and Table 2).

### 3.5. Methodological Quality Assessment within Studies and Overall Quality Assessment across Studies

Evaluating the methodological quality of the seven included studies, we found low (*n* = 1) to moderate (*n* = 3) and high (*n* = 3) quality (Table 1). In Figure 3, we show an overview of quality ratings within (Panel A) and across studies (Panel B), highlighting areas with higher or lower risk ratings.

We detected bias predominantly in the domains of sample size justification (selection bias) and blinded assessment (detection bias) [all 7 (100%) studies were related with a high risk of bias] (Figure 3, Panel B). Six studies out of seven (85.7%) were associated with a higher risk of bias regarding the different levels of exposure rate and four studies out of seven (57.1%) were related to a prevalent risk of bias in confounding and exposure measures (Figure 3, Panel B). We judged the GRADE overall certainty of evidence as moderate for the association of BMS with anxiety disorders and low for the association of BMS with depressive disorders (Table 2).

## 4. Discussion

In the present systematic review, in middle-aged and older subjects, anxiety disorders were found to be the most common psychiatric disorders related to BMS (63.7%), followed by depressive disorders (36.3%). In fact, in these age groups, we found a moderate association of BMS with anxiety disorders and a low association of BMS with depressive disorders. The role of pain appeared to be controversial in explaining these associations. Furthermore, also in middle and older age, females showed higher risk of developing BMS than males, and there was a worsening of symptoms such as sleep disorders, personality traits, and biopsychosocial changes in addition to anxiety and depression in subjects with BMS, as suggested by the present study-specific findings.

In a recent systematic review and meta-analysis without selection for age groups, all selected studies but one showed some evidence of the association between psychological factors and BMS [11]. This study confirmed the present findings in middle and older age, suggesting that among BMS patients, anxiety and depression were the most common and most frequently studied psychiatric disorders [11]. The present findings were confirmed also by a very recent systematic review without selection for age groups showing a link of BMS with psychiatric disorders, particularly anxiety and/or depressive symptoms [12].

In fact, several studies have highlighted the correlation of anxiety disorders with BMS; however, most of these studies showed cross-sectional findings and cannot show the time sequence absolutely. Hence, due to the nature of such studies, the causative relationship between anxiety disorders and BMS cannot be clearly established. In anxiety- associated BMS, many endocrine and metabolic changes may occur [6]. Among these changes, among physiological effects, there is a rise in cortisol levels [16], and possible neuropathic mechanisms of BMS should be further investigated among different pathogeneses of this syndrome [16]. This hypothesis was supported by a case-control study showing an interplay among high anxiety levels, salivary cortisol levels, and BMS [17].

Moreover, in BMS subjects, anxiety disorder may determine a secondary depression and depressive symptoms could contribute to pain, suggesting that pain could be a somatic feature of a depressive disorder [20]. In the present study, depressive and anxiety disorders in subjects with BMS did not appear to be associated with and aggravated by pain intensity in middle and older age. In 2015, a study investigated the association between pain and psychosocial characteristics in middle-aged (45–64 years) and older (65–84 years) subjects presenting BMS and temporomandibular disorders (TMDs) [22]. These disorders showed different pain intensities, with increasing symptoms related to advancing age also associated with differences between TMDs and BMS, thus resulting in different psychosocial factors, suggesting that BMS could have a different pathophysiological etiology in relation to a patient’s age [22], according to similar studies on TMDs [23]. Moreover, stress is an arousal state in response to environmental stressors and may be characterized by biopsychosocial changes with a positive or negative nature, i.e., BMS [24], and physical activity may be an activator of pain-inhibitory systems, so reducing the severity of pain in long-lasting pain conditions [25]. According to this hypothesis, and given that currently little is known on the role of physical activity in women with BMS, a study suggested that in 56 women with BMS, perceived stress was higher and weekly physical activity was reduced compared with controls [21]. Of note, BMS may be also related to chronic pain outside of the orofacial sphere. In fact, in subjects with BMS, other pain symptoms involving similar mechanisms can be found, especially vulvodynia in women and penoscrotodynia in men as well as pudendal neuralgia, grouped together using the term “pelvodynia” [26].

Furthermore, different findings suggested that a major theme in clinical research on BMS should be the relationship between the disorder and personality traits [21,27,28]. In fact, this hypothesis was supported by a study in which subjects with BMS presented a characteristic personality trait that made them more likely to be cautious [27]. In addition, they also had low self-directedness tendencies and high harm avoidance, traits probably associated with depressive disorders [26]. Furthermore, BMS subjects showed higher rates of obsessive–compulsive, schizotypal, and paranoid personality disorders [28], confirming some abnormal personality traits previously associated with BMS, i.e., obsessions and compulsions, personal sensitivity, lower socialization capability, and emotional repression.

The present study-specific findings suggested that, in addition to anxiety and depression, in middle-aged and older subjects with BMS, there was a worsening of sleep disorders, personality traits, and biopsychosocial changes. However, current findings on sleep quality, pain, and depressive symptoms are based on self-assessment, which could generate some bias. However, subjects with primary BMS presented a significant decrease in sleep quality, so confirming the comorbidity between sleep and depressive disorders, confirming the central role of sleep evaluation in assessing BMS and generating viable options for improving its treatment [29]. Finally, altered estrogen levels may contribute to an increased risk of BMS in females [6]. These findings are supported by the fact that estrogen receptors have been found not only in the vaginal mucosa but also in the salivary glands in the tongue. One cohort study also demonstrated that postmenopausal patients who received hormone replacement therapy experienced relief from their BMS symptoms [30]. Consistent with these findings, we observed that the risk of incident BMS events was significantly increased in female patients with depression and anxiety

Therapeutic strategies for subjects with BMS include local and systemic treatments [31]. Several local treatments have been used in subjects with BMS with varying degrees of success. In BMS, mouthwashes with clonazepam may be effective [26], and desensitization to capsaicin using mouthwashes containing hot peppers diluted in water has been also recommended. Systemic treatments with gabapentin (1200 mg/day for over 26–32 weeks) and pregabalin (50–150 mg/day) showed efficacy in subjects with BMS [31,32]. Other systemic treatments that succeed in BMS are dosulepin (75 mg/day), α-lipoic acid (800 mg/day), duloxetine (20–60 mg/day), and clonazepam (3 mg/day) [26,31,33,34] and the combination of gabapentin and nortriptyline, gabapentin and α-lipoic acid, and diazepam and olanzapine [31,34].

We must acknowledge some limitations of the present study. The main limitation of the present systematic review was a lack of evidence-based clinical studies in the two areas (anxiety and depression) investigated for BMS in these age groups. In fact, we found only seven reports to be included. In addition, the GRADE assessment reported data with a moderate/low strength of evidence. Finally, the present was a systematic review study, and we could not directly examine the pathological mechanisms underlying the relationship between the BMS condition and depressive and anxiety disorders in middle-aged and older adults. The findings of the present study could reflect the wide variations of clinical presentation of BMS in middle and older age; however, this topic should be evaluated in further case-control and population-based studies designed in particular for these age groups.

## 5. Conclusions

In conclusion, evaluating the findings of the selected studies, an important objective in personalized medicine, could be early identification and recognition of the association of BMS with depressive and anxiety disorders in middle-aged and older adults to frame the affected subjects who would be directed toward a treatment that is not only of dental relevance. In fact, the present findings confirmed the central role of an approach guided by a multidisciplinary team with strict collaboration between dentists, psychiatrists, neurologists, and psychologists to evaluate multimorbidity and for personalized treatment of subjects with BMS. Finally, in these age groups, there was a worsening of sleep disorders, personality traits, and biopsychosocial changes, in addition to anxiety and depressive disorders that may reduce the ability to adapt to patients with BMS. In the next future, we should implement larger prospective studies, with a population-based design and longer follow-up periods, conducted in different countries/geographical areas to evaluate the association between BMS with depressive and anxiety disorders in middle-aged and older adults to increase the low strength of evidence of the collected findings, also addressing potential bias and confounding sources.

## Figures and Tables

**Figure 1 jpm-13-01014-f001:**
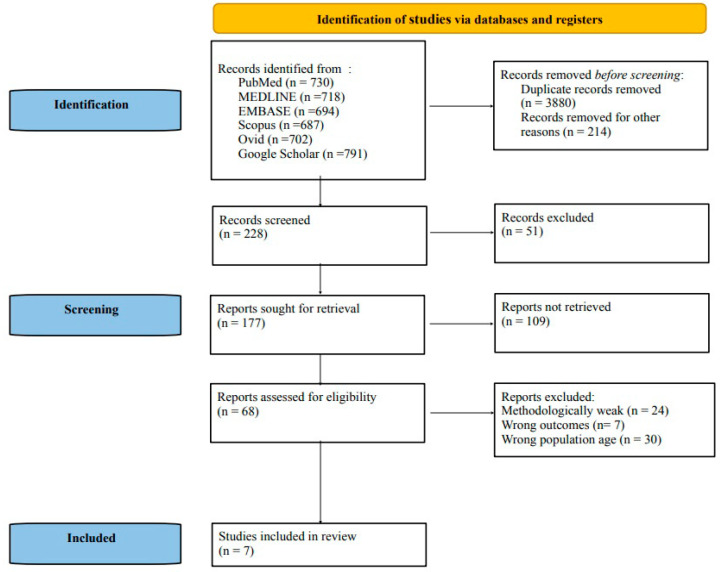
Preferred Reporting Items for Systematic Reviews and Meta-analyses (PRISMA) 2020 flow chart.

**Figure 2 jpm-13-01014-f002:**
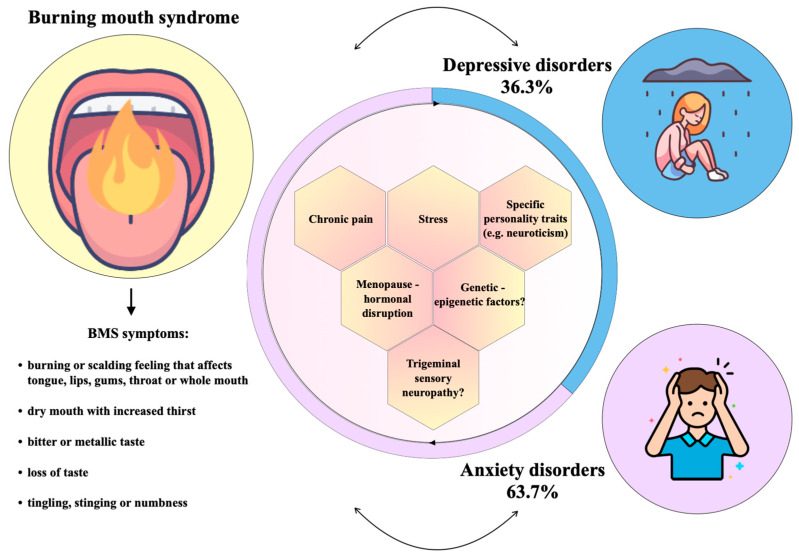
Burning mouth syndrome associated with anxiety and depressive disorders, investigated in the selected studies, with their percentage distribution.

**Figure 3 jpm-13-01014-f003:**
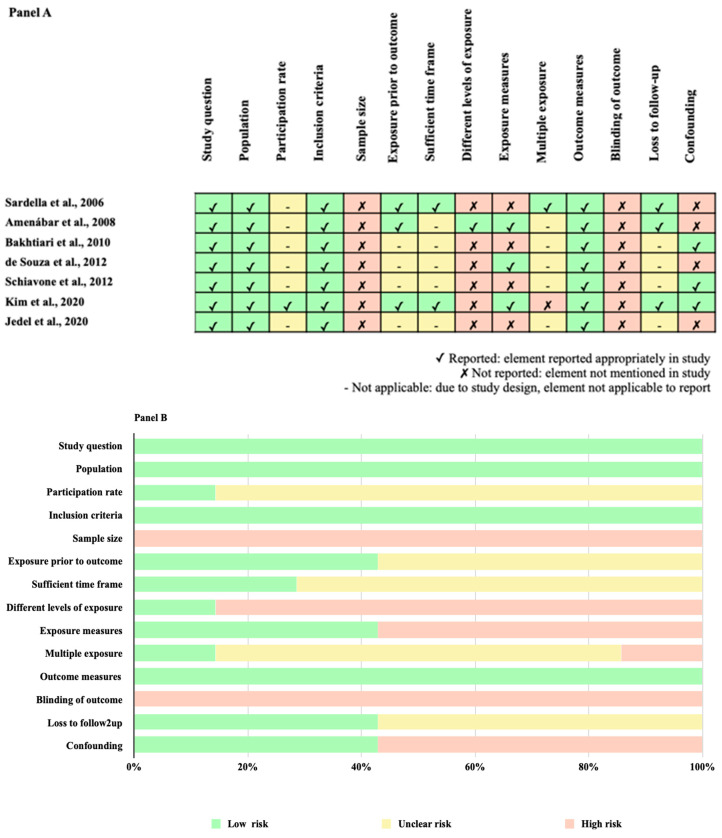
Methodological quality assessment within studies (Panel **A**) [1,16,17,18,19,20,21] and overall quality assessment across studies (Panel **B**).

**Table 1 jpm-13-01014-t001:** Selected studies investigating burning mouth syndrome (BMS) and depressive and anxiety disorders as outcomes (*n* = 7) and quality appraisal summary.

Authors, Year [Reference]	Outcome(s)	Outcome(s) Assessment Tool(s) and Pain Assessment Tools	Design (Follow-Up)	N	Age	Sex	Setting(s)	Country	Quality Assessment	Main Findings
Sardella et al., 2006 [16]	Depressive disorders Anxiety disorders	Hospital Anxiety and Depression Scale VAS	Longitudinal Case-control (3 years)	67/54	62.3 years (4.6)/56.8 years (5)	15% M, 85% F/10% M, 90% F	Hospital (outpatients)	Europe (Italy)	High	This prospective case-control study showed significant differences between BMS and control subjects regarding anxiety and depressive disorders.
Amenábar et al., 2008 [17]	Anxiety disorders	Beck Anxiety Inventory	Longitudinal Case-control (N/A)	30/30	61.6 years (10.7)/63.2 years (9.7)	20% M, 80% F/N/A	Hospital (outpatients)	South America (Brazil)	High	BMS was positively associated with higher anxiety and elevated salivary cortisol levels.
Bakhtiari et al., 2010 [18]	Anxiety disorders	Cattell Anxiety Scale VAS	Cross-sectional Case-control	50/50	60 years+ 70.3 years (9.9)/70.9 years (9.8)	12% M, 88% F/22% M, 78% F	Hospital (outpatients)	Asia (Iran)	Moderate	This cross-sectional study suggested that both state and trait anxiety were related to the presence of BMS.
de Souza et al., 2012 [19]	Depressive disorders Anxiety disorders	The Mini-International Neuropsychiatric Interview-Plus (MINI-Plus) Hamilton Rating Scale for Depression Beck Depression Inventory State-Trait Anxiety Inventory VAS	Cross-sectional Case-control	30/31	63.8 years (11.8)/63.8 years (11.8)	3.3% M, 97.7% F/3.2% M, 97.8% F	Hospital (outpatients)	South America (Brazil)	Moderate	The findings of this cross-sectional study confirmed clinical observations on the fact that subjects with BMS may have a particular psychiatric and/or psychological profile.
Schiavone et al., 2012 [20]	Depressive disorders Anxiety disorders	Hamilton Rating Scale for Depression State-Trait Anxiety Inventory Form Y 1–2 Symptom Checklist-90-Revised VAS	Cross-sectional Case-control	53/51	55.26 years (11.50)/54.02 years (13.28)	30.2% M, 69.8 F/33.3% M, 66.7% F	Hospital (outpatients)	Europe (Italy)	Moderate	This cross-sectional study highlighted that in BMS there were psychiatric symptoms (anxiety and depression) with a possible association with pain.
Kim et al., 2020 [1]	Depressive disorders Anxiety disorders	Korean Standard Classification of Diseases, Sixth Revision	Retrospective Case-control (10 years)	695/362	45 years+	38.6% M, 61.4% F/38.1% M 61.9% F	Hospital (inpatients and outpatients)	Asia (South Korea)	High	Findings of this observational study suggested that BMS was associated with increased incidence of depression and anxiety but not of dementia or Parkinson’s disease.
Jedel et al., 2020 [21]	Anxiety disorders	Swedish universities Scales of Personality VAS	Cross-sectional Case-control	56/56	67.8 years (8.9)/67.8 years (8.9)	(100% F/N/A)	Hospital (outpatients)	Europe (Sweden)	Low	SSP subscales Somatic Trait Anxiety, Psychic Trait Anxiety, Stress Susceptibility, and Verbal Trait Aggression differed between women with BMS and controls, and the personality factor scores for neuroticism and aggressiveness were higher.

M: males; F: females; N/A: not available; VAS: Visual Analogue Scale; SSP: Swedish universities Scales of Personality.

**Table 2 jpm-13-01014-t002:** Summary of findings on psychiatric disorders associated with burning mouth syndrome (BMS) in middle-aged and older adults.

Psychiatric Disorders	Evidence Base	Strength of Association	Strength of Evidence (GRADE)	Comments
Depressive disorders [1,16,19,20]	Four studies *n* = 1.343	BMS vs. depression (45–64 years): HR = 1.45, 95% CI = 1.03–2.04; BMS vs. depression (64 years+): HR = 1.68, 95% CI = 1.16–2.43 [1] BMS vs. depression: OR = 3.833, 95% CI = 1.528–9.572 [16] BMS vs. current major depressive disorder (BMS group vs. control group): p = 0.004 (Chi-square test); BMS vs. past major depressive disorder (BMS group vs. control group): p = 0.006 (Chi-square test) [19] BMS vs. depression (SCL-90-R) (BMS group vs. control group): p =< 0.001 (ANOVA); BMS vs. depression (HAM-D) (BMS group vs. control group): p =< 0.001 (ANOVA) [20]	⊕⊕ Low	Low association of BMS with depressive disorders, with estimates partly provided; a few studies included but with a large sample size.
Anxiety disorders [1,16,17,18,19,20,21]	Seven studies *n* = 1.615	BMS vs. anxiety (45–64 years): HR 1.72, 95% CI 1.30–2.29; BMS vs. anxiety (64 years+): HR 2.13, 95% CI 1.57–2.91 [1] BMS vs. anxiety (HAD Scale): OR 4.256, 95% CI 1.780–10.148 [16] BMS vs. anxiety (BMS group vs. control group): p = 0.001 (Fisher exact test) [17] BMS vs. anxiety: r = 0.431, p < 0.001 [18] BMS vs. generalized anxiety disorder (BMS group vs. control group): p = 0.012 (Chi-square test) [19] BMS vs. anxiety (SCL-90-R), (BMS group vs. control group): p = 0.002 (ANOVA); BMS vs. anxiety (STAI Y1), (BMS group vs. control group): p = 0.026 (ANOVA); BMS vs. anxiety (STAI Y2), (BMS group vs. control group): p = 0.046 (ANOVA) [20] BMS vs. somatic trait anxiety (BMS group vs. control group): (p < 0.001) (Wilcoxon sign rank test) [21]	⊕⊕⊕ Moderate	Moderate association of BMS with anxiety disorders, with estimates partly provided; multiple studies included, with also a large sample size.

OR: odds ratio; HR: hazard ratio; CI: confidence interval; SCL-90-R: Symptom Checklist-90-Revised; HAM-D: Hamilton Rating Scale for Depression; HAD Scale: Hospital Anxiety and Depression Scale; STAI Y1: State-Trait Anxiety Inventory Form Y 1; STAI Y2: State-Trait Anxiety Inventory Form Y 2.

## Data Availability

Not applicable.

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
