# Peer review of "Exploring the Association of Burning Mouth Syndrome with Depressive and Anxiety Disorders in Middle-Aged and Older Adults: A Systematic Review"

_jpm, 2023, doi:10.3390/jpm13061014_

Round 1

Reviewer 1 Report

The research makes an contribution to the literature. After taking into account a few corrections, the work does not raise any objections.

Introduction

The following should be supplemented (due to the non-specific nature of the ailments, there is no information on the clinical picture ....

Materials and Methods

Inclusion/exclusion criteria - should be clearly stated.

Results

Tabela 1. Quality  Assessment - unclear (of what?) Burning mouth syndrome (BMS) is an idiopathic chronic pain disorder characterized.

No information on pain assessment.

Discussion

To be supplemented.

Author Response

Reviewer 1
The research makes an contribution to the literature. After taking into account a few corrections, the work does not raise any objections.

Introduction
1. The following should be supplemented (due to the non-specific nature of the ailments, there is no information on the clinical picture ....
1. We thank this Reviewer for his/her thoughtful comment. We included it in this revised version of the manuscript (please, see page 2).

Materials and Methods
2. Inclusion/exclusion criteria - should be clearly stated.
2. In this revised version of the manuscript, we included inclusion/exclusion criteria (please, see page 3).

Results
3. Tabela 1. Quality Assessment - unclear (of what?) Burning mouth syndrome (BMS) is an idiopathic chronic pain disorder characterized. No information on pain assessment.
3. In Table 1, we showed a synopsis of the principal findings of included studies investigating burning mouth syndrome and depressive and anxiety disorders as outcomes (N=7) with also a quality appraisal summary. We described the assessment tools of the outcomes (depressive and anxiety disorders). In this revised version of the manuscript, we included pain assessment tools when reported.

Discussion
4. To be supplemented.
4. In the Discussion section, in this revised version of the manuscript, we included some amount of discussion on the therapeutic strategies for subjects with BMS including local and systemic treatments (please, see page 14).

Reviewer 2 Report

This is an interesting and good paper.The Burning Mouth Syndrome is a difficult problem.The meta-analysis of the Syndrome is important.

What I miss from the paper is that it doesen'twrite about the treatment.It's a complex problem,but a lot can be done to help,especially in terms of treating xerostomia. 

I suggest publishing the paper after supplementing it with therapy.

Author Response

Reviewer 2
This is an interesting and good paper. The Burning Mouth Syndrome is a difficult problem. The meta-analysis of the Syndrome is important.

1. What I miss from the paper is that it doesen't write about the treatment. It's a complex problem, but a lot can be done to help, especially in terms of treating xerostomia. I suggest publishing the paper after supplementing it with therapy.
1. We thank this Reviewer for his/her thoughtful comment. In the Discussion section, in this revised version of the manuscript, we included some amount of discussion on the therapeutic strategies for subjects with BMS including local and systemic treatments (please, see page 14).